# CONNECT LATER: IMPROVING FINE-TUNING FOR ROBUSTNESS WITH TARGETED AUGMENTATIONS

## ABSTRACT

Models trained on a labeled source domain (e.g., bright, nearby astronomical objects) often generalize poorly when deployed on an out-of-distribution (OOD) target domain (e.g., faint, distant objects). In the domain adaptation setting where unlabeled target data is available, self-supervised pretraining (e.g., masked autoencoding or contrastive learning) is a promising method to mitigate this performance drop. Pretraining improves OOD error when the generic data augmentations used (e.g., masking or cropping) connect the source and target domains, which may be far apart in the input space. In this paper, we show on real-world tasks that standard fine-tuning after pretraining does not consistently improve OOD error over just supervised learning on labeled source data. To better leverage pretraining for distribution shifts, we propose Connect Later: after pretraining with generic augmentations to learn good representations within the source and target domains, fine-tune with *targeted augmentations* designed with knowledge of the distribution shift to better connect the domains. Connect Later improves average OOD error over standard fine-tuning and supervised learning with targeted augmentations on 4 real-world datasets: astronomical time-series classification (ASTROCLASSIFICATION) by 12%, redshift prediction for astronomical time-series (REDSHIFTS) by 0.03 RMSE (11% relative), wildlife species identification (IWILDCAM-WILDS) by 0.8%, and tumor detection (CAMELYON17-WILDS) by 1.1%, achieving the state-of-the-art on ASTROCLASSIFICATION, IWILDCAM-WILDS with ResNet-50, and CAMELYON17-WILDS with DenseNet121.

## 1 INTRODUCTION

In many real-world scenarios, machine learning models are deployed on data that differ significantly from the training data (Quiñonero-Candela et al., 2009; Koh et al., 2021). We focus on unsupervised domain adaptation (Shimodaira, 2000; Blitzer et al., 2006; Sugiyama et al., 2007), where we have labeled data from a source domain and unlabeled data from a target domain. We aim to learn a model that generalizes well to these out-of-distribution (OOD) target domain inputs. A real-world example is in astronomy, where machine learning is used to predict properties of astronomical objects from telescope data. These properties are key to understanding the physical processes of the universe, including supernovae, black holes, and cosmic expansion (Boone, 2019; Lin & Pandya, 2020). However, ground truth labels for these tasks require expert labeling, which is only feasible for a small subset of bright, nearby objects (LSST Science Collaboration et al., 2009). As a result, the labeled data is not representative of the full unlabeled dataset of observations, which contains mostly faint, distant objects.

Self-supervised pretraining on unlabeled data has shown promising results on real-world problems (Caron et al., 2020; Shen et al., 2022; Devlin et al., 2019; Radford et al., 2021; Sagawa et al., 2022). In contrast to traditional domain adaptation methods in deep learning that focus on learning domain-invariant features(Ganin et al., 2016; Kang et al., 2019; Tzeng et al., 2017; Saenko et al., 2010; Sun et al., 2016; Hoffman et al., 2018), pretraining learns representations that are not domain-invariant, but instead decompose the class and domain information, facilitating transfer across domains (Shen et al., 2022). A favorable decomposition depends on data augmentation to connect the source and target domains and learn transferable representations. Intuitively, strongly augmented (e.g. masked or cropped) source and target inputs are more likely to look similar if they are from the same class (e.g., cropping out the face of a lion in different habitats) than from different classes (e.g., no body parts of elephants and lions are alike). These generic augmentations must be strong enough to connect the domains with the unlabeled data without knowledge of the distribution shift.

In this paper, we find on real-world benchmarks that standard fine-tuning after pretraining does not consistently improve OOD error over just supervised learning with labeled source data (Section 3). On the other hand, supervised learning with *targeted augmentations* (Gao et al., 2023) designed for the distribution shift consistently improves OOD error over the supervised learning baseline. Our observations show that

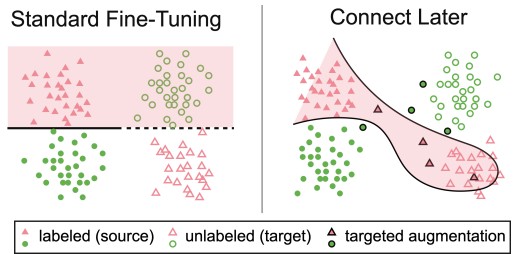

Figure 1: Overview of the Connect Later framework for domain adaptation applied to a toy binary classification problem with two domains (filled and unfilled points), showing the representations in $\mathbb{R}^2$. **(Left)** After contrastive pretraining with generic augmentations, the classes within each domain are linearly separable in representation space. Since the domains are far apart in input space, generic augmentations may not connect the domains, resulting in misalignment in the pretrained representations. In this case, a classifier (with a linearly extrapolating decision boundary, dashed and solid line) learned on labeled source data will misclassify the target data. **(Right)** Connect Later employs targeted augmentations (filled points with black border), which are designed with knowledge of the distribution shift, to connect the domains better, resulting in a classifier that generalizes well to the target domain.

pretraining with generic augmentations is not sufficient to learn transferable representations for all distribution shifts. Can pretraining still be used to improve OOD error in this case?

To better leverage pretraining for domain adaptation, we propose the Connect Later framework (Figure 1): after pretraining with generic augmentations, fine-tune with targeted augmentations (Section 4). Pretraining learns good representations within each domain, while targeted augmentations better connect the domains. We also provide a general methodology for constructing these targeted augmentations, where the augmented inputs match the target distribution on a feature space where the domains differ.

We evaluate our framework on 4 real-world datasets (Section 5): wildlife identification (IWILDCAM-WILDS, Beery et al., 2020; Sagawa et al., 2022) tumor detection (CAMELYON17-WILDS, Bandi et al., 2018; Sagawa et al., 2022), and 2 astronomical time series tasks, ASTROCLASSIFICATION and REDSHIFTS, which we curate from The PLAsTiCC team et al. (2018). In Section 6, we show that Connect Later improves ID and OOD performance over standard fine-tuning or supervised learning with targeted augmentations across all datasets. Connect Later achieves the state-of-the-art on three benchmarks, improving OOD accuracy on ASTROCLASSIFICATION by 3% (Boone, 2019), IWILDCAM-WILDS with ResNet-50 by 0.8%, and CAMELYON17-WILDS with DenseNet121 by 1.1%. We also contribute the REDSHIFTS dataset, on which Connect Later produces 11% and 14% relative improvement over standard fine-tuning and supervised learning with targeted augmentations, respectively.

## 2 SETUP

We consider a prediction problem from an input space $\mathcal{X}$ to a label space $\mathcal{Y}$, where $\mathcal{Y} = \{1,...,k\}$ for classification and $\mathcal{Y} \in \mathbb{R}$ for regression.

**Domain adaptation.** Let $P_S$ and $P_T$ be the source and target input distributions over $\mathcal{X}$, respectively. We consider unsupervised domain adaptation, where we have access to source inputs $x \sim P_S$, with corresponding labels $y \in \mathcal{Y}$ sampled from the label distribution $p^*(\cdot \,|\, x)$, along with unlabeled target inputs $x \sim P_T$. Let the unlabeled distribution $P_U = \beta P_S + (1-\beta)P_T$ be a mixture of the source and target, where $\beta \in [0,1]$. In some practical cases, $P_U$ may also be a broader unlabeled distribution. The goal is to learn a model $f_\theta : \mathcal{X} \to \mathcal{Y}$ that minimizes error on the target domain $L_T(f) = \mathbb{E}_{x \sim P_T, y \sim p^*(\cdot \,|\, x)}[\text{loss}(f(x), y)]$. For example, $\text{loss} : \mathcal{Y} \times \mathcal{Y} \to \mathbb{R}$ is the 0-1 loss in classification and squared loss in regression.

**Augmentations.** Augmented inputs $x'$ are drawn from an augmentation distribution $\mathcal{A}(\cdot | x)$. Training with augmented inputs is often used to improve the robustness of the model (Hendrycks et al., 2019; 2020) as well as to provide an objective for self-supervised pretraining (Caron et al., 2020; Shen et al., 2022). In this work, we define two distinct augmentation distributions, $\mathcal{A}_{\text{pre}}$ and $\mathcal{A}_{\text{ft}}$, for the pretraining and fine-tuning steps, respectively.

**Pretraining for domain adaptation.** Pretraining for domain adaptation consists of two steps: self-supervised pretraining on unlabeled data, then supervised fine-tuning on labeled source data (Shen et al.,

2022). For simplicity below, we consider the population objectives. During the pretraining step, we optimize model parameters $\theta$ with the pretraining objective

$$L_{\text{pretrain}}(\theta) = \mathbb{E}_{B \sim P_U^m}[\text{loss}_{\text{pretrain}}(B, \mathcal{A}_{\text{pre}}; \theta)], \tag{1}$$

where $B$ is a batch of $m$ inputs. The pretraining loss $\text{loss}_{\text{pretrain}}$ encompasses both masked autoencoding, which operates on a single example, and contrastive learning, which operates on a batch. The output of pretraining is a set of pretrained parameters $\theta_{\text{pre}}$.

Fine-tuning then uses labeled source data to adapt the parameters, initialized with the pretrained parameters $\theta_{\text{pre}}$, to a specific downstream task with the objective

$$L_{\text{ft}}(\theta) = \mathbb{E}_{x \sim P_S, y \sim p^*(\cdot|x), x' \sim \mathcal{A}_{\text{ft}}(\cdot|x)}[\text{loss}_{\text{ft}}(x', y; \theta)] \tag{2}$$

where $\text{loss}_{\text{ft}}$ is a fine-tuning objective such as softmax cross entropy loss for classification or squared error for regression.

Pretraining has been shown to improve OOD performance in both vision and natural language (e.g., Han & Eisenstein, 2019; Radford et al., 2021; Shen et al., 2022). Typically, the pretraining augmentations $\mathcal{A}_{\text{pre}}$ are generic transformations, such as random cropping in vision or masking in NLP (Caron et al., 2020; Chen et al., 2020; He et al., 2020; Radford et al., 2021; Shen et al., 2022; He et al., 2022; Devlin et al., 2019). Fine-tuning augmentations $\mathcal{A}_{\text{ft}}$ have not been studied extensively and are typically also generic or simply the identity transformation (Sagawa et al., 2022; Devlin et al., 2019).

**Standard fine-tuning.** We refer to **standard fine-tuning** as the pretraining+fine-tuning procedure where $\mathcal{A}_{\text{ft}}(x' \mid x) = 1$ if $x' = x$ (no fine-tuning augmentations). In our experiments, the standard fine-tuning procedure is linear probing then fine-tuning (LP-FT) (Kumar et al., 2022), which has been shown to improve ID and OOD performance over vanilla fine-tuning. In LP-FT, we first learn a linear predictor on top of frozen pretrained features before fine-tuning all the parameters jointly.

**ERM with augmentations.** As a baseline, we consider empirical risk minimization (ERM) with data augmentation, which optimizes the fine-tuning objective (Equation 2) on labeled source data with randomly initialized parameters. In this paper, we refer to **ERM** as the instantiation where $\mathcal{A}_{\text{ft}}(x' \mid x) = 1$ if $x' = x$ (no augmentations) and **ERM + targeted augmentations** as the instantiation with $\mathcal{A}_{\text{ft}}$ that is designed with knowledge of the distribution shift.

## 3   PRETRAINING PRODUCES INCONSISTENT OOD PERFORMANCE

We compare ERM, ERM+targeted augmentations, and standard fine-tuning on 4 real-world datasets: ASTROCLASSIFICATION, REDSHIFTS, IWILDCAM-WILDS, and CAMELYON17-WILDS. For pretraining, we use SWaV contrastive learning (Caron et al., 2020) with cropping at multiple resolutions for IWILDCAM-WILDS and CAMELYON17-WILDS, and masked autoencoding with 60% of observations masked for ASTROCLASSIFICATION and REDSHIFTS.

Table 1 shows that standard fine-tuning after pretraining with strong generic augmentations does not produce consistent OOD performance. Standard fine-tuning in IWILDCAM-WILDS does not improve either ID or OOD performance over ERM, showing a 0.7% lower ID performance and comparable OOD performance. However, pretraining is clearly beneficial in ASTROCLASSIFICATION, where standard fine-tuning improves both ID and OOD accuracy by 6-7% over ERM. We hypothesize that the generic pretraining augmentations connect the domains better for some tasks and distribution shifts than others.

In Appendix D, we provide a simple binary classification example of when contrastive pretraining fails for domain adaptation, following a similar augmentation graph construction to Shen et al. (2022). In this graph, the nodes are examples from different class-domain pairs, and edge weights are defined by "connectivity": the probability that the pretraining augmentation transforms an example from one class-domain pair to another. When the connectivity structure misaligns the source and target domains, such that examples from the same class are not more "connected" than examples from different classes across the domains, a linear classifier trained on these pretrained representations will not transfer from source to target. This could happen, for example, when the source and target are far apart in input space and the connectivity is low between all example pairs from different domains. However, the representations are still useful since the classes are linearly separable within each domain. How do we leverage these pretrained representations when they may not transfer well across domains?

## 4   CONNECT LATER: PRETRAIN FIRST, TARGETED AUGMENTATIONS LATER

In this work, we propose the Connect Later framework (Figure 1):

|  | AstroClassification | | iWildCam | |
| --- | --- | --- | --- | --- |
|  | ID Test Acc | OOD Acc | ID Test Macro F1 | OOD Test Macro F1 |
| ERM | $71.59 \pm 1.10$ | $61.26 \pm 1.10$ | **46.4 $\pm$ 0.5** | $30.4 \pm 0.6$ |
| Standard fine-tuning | **78.84 $\pm$ 0.97** | **67.84 $\pm$ 0.70** | **46.4 $\pm$ 0.8** | **31.2 $\pm$ 0.6** |

Table 1: Standard fine-tuning produces substantial gains in ID and OOD performance on ASTROCLAS-SIFICATION compared to an ERM baseline, but IWILDCAM-WILDS does not benefit from pretraining. Results are averaged over 5 trials and show standard deviations for ASTROCLASSIFICATION, and 15 trials and show 95% confidence intervals for IWILDCAM-WILDS. Rows with means within 1 interval of the best mean are shown in bold.

1. Pretrain on unlabeled data with generic augmentations as in Equation 1, producing pretrained parameters $\theta_{\text{pre}}$. This step learns good representations of the source and target domains and allows us to reuse the pretrained model for multiple downstream tasks.
2. Design a targeted augmentation $\mathcal{A}_{\text{ft}}$ (discussed below) and fine-tune the pretrained model, initializing from $\theta_{\text{pre}}$, as in Equation 2. The targeted augmentation better connects the domains for the distribution shift.

**Designing targeted augmentations.** How do we design these targeted augmentations? We provide a general methodology based on matching the target distribution on a feature space:

1. Identify a feature space $\mathcal{Z}$. We assume that we can label $z \in \mathcal{Z}$ for each input and that the source and target domains largely differ on this feature space. One such example is the space of spurious, domain-dependent features (e.g., camera angle or resolution for IWILDCAM-WILDS), which is the approach followed by Gao et al. (2023).
2. Fit a transformed feature distribution $\hat{p_T}(z'|z)$ to the target feature distribution.
3. Create a transformation distribution $T(x'|x,z')$ where $x'$ is the augmented version of $x$ with $z = z'$. In this paper, we define $T$ with domain knowledge.
4. Given an input $x$, generate augmentations by sampling a new feature $z'$ from $\hat{p_T}(z' \mid z)$, then sampling an augmentation from $T(x'|x,z')$. The resulting targeted augmentation probabilities are $\mathcal{A}_{\text{ft}}(x'|x) = \sum_{z'} T(x'|x,z') \hat{p_T}(z'|z)$.

**Targeted augmentation example.** We follow the procedure outlined above to design a targeted augmentation for ASTROCLASSIFICATION and REDSHIFTS (see Appendix B.2 for details). Recall that in these datasets, expert labels are only available for bright, nearby objects, while the unlabeled dataset contains mostly faint, distant objects. Nearby objects have lower redshift values than distant objects, causing the source and target redshift distributions to be mismatched (Appendix Figure 4).

1. The source and target domains primarily differ on their redshift distributions, so we identify this scalar feature as $z$.
2. We roughly fit the target redshift distribution while constraining the transformed redshift value to not be too far from the original redshift $z$, such that $\hat{p_T}(z' \mid z)$ is distributed as loguniform$(0.95z, \min(1.5(1+z)-1, 5z))$, following Boone (2019).
3. We define a transformation distribution $T(x'|x,z')$, where $x$ is a time-series of flux values at multiple wavelengths and $z'$ is a new redshift value to transform to. We first fit a Gaussian process that models $x$ as a function of time and wavelength. Given $z'$, we rescale the timestamps and wavelengths of the original input to account for the physical effects of the new redshift value. Then, we sample $\tilde{x}'$ from the Gaussian process at these new timestamps and wavelengths. Finally, we produce the transformed input $x'$ by scaling the flux values to account for $z'$.
4. We sample $z'$ from $\hat{p_T}(z'|z)$ and then sample augmentations $x'$ from $T(x'|x,z')$.

**Simple example where Connect Later achieves 0 OOD error.** In our simple binary classification example in Appendix D, we show that when the connectivity structure is misaligned, both standard fine-tuning with contrastive pretraining and ERM + targeted augmentations have high OOD error, while Connect Later achieves 0 OOD error. In this setting, ERM with targeted augmentations is unable to achieve 0 OOD error since some target inputs are "unreachable" via targeted augmentations of source inputs. The pretraining step in Connect Later uses unlabeled target data to learn representations where label information from source data can propagate to all target inputs.

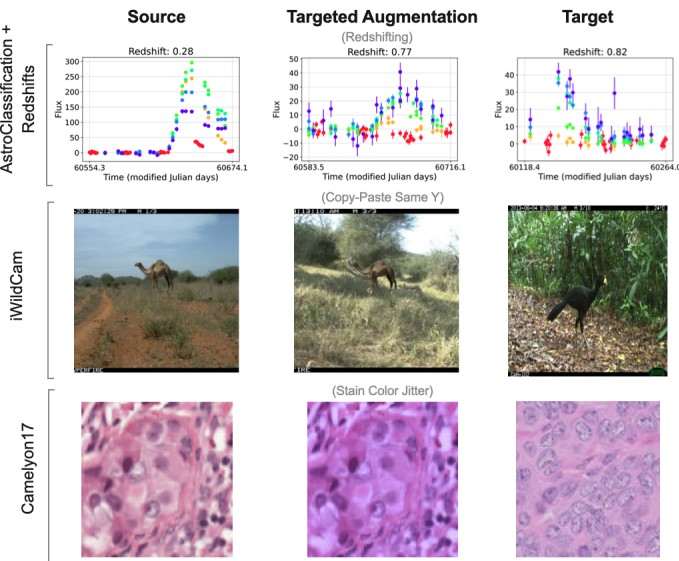

Figure 2: An example from the source dataset (left), an augmented version of the source example (middle), and an example from the target dataset (right) for our 3 tasks. (**Top row**) The target dataset in ASTRO-CLASSIFICATION and REDSHIFTS is much higher redshift than the source dataset. We apply the redshifting augmentation to simulate placing source objects at a higher redshift to better match the target dataset. The flux errors and flux values of the augmented example (middle) show much better resemblance of the target example. (**Middle row**) The IWILDCAM-WILDS target dataset comes from unseen cameras placed in potentially new habitats, so we randomize the habitat background by applying the Copy-Paste Same Y augmentation. This algorithm places source dataset animals into empty backgrounds from other cameras that have observed the same species. (**Bottom row**) The CAMELYON17-WILDS target dataset comes from unseen hospitals. We apply the Stain Color Jitter augmentation to simulate a different staining procedure that may be used by other hospitals. IWILDCAM-WILDS and CAMELYON17-WILDS image examples are from Gao et al. (2023).

## 5 DATASETS

We consider 3 real-world tasks using astronomical time-series and wildlife camera trap images. We show examples from the source, augmented, and target datasets in Figure 2.

### 5.1 TASKS

**Astronomical object classification (ASTROCLASSIFICATION).** Astronomical object classification (Boone, 2019; Allam Jr. & McEwen, 2022) involves predicting the object type (e.g., type II supernova) from a time series of an object's brightness at multiple wavelengths (*light curves*). We curate this dataset from the Photometric LSST Astronomical Time Series Classification Challenge (PLAsTiCC, The PLAsTiCC team et al., 2018) (details in Appendix A.1).

- **Source:** Time-series of bright, nearby objects with expert labels
- **Target:** Time-series of all observed objects from the telescope, often faint and distant (higher redshift). Follow-up observation, which is required for expert labeling, is too expensive for these objects.
- **Targeted Augmentation:** We augment the labeled dataset by redshifting each object, i.e., simulating its observed properties as if it were further away (Section 4).
- **Task:** 14-class astronomical object classification

**Redshift regression (REDSHIFTS).** Similar to object type, redshift information is also available only for bright, nearby objects. We predict the scalar redshift value of each object and minimize mean squared error. This task has been studied for individual object types, such as quasars (Nakoneczny et al., 2021) and type Ia supernovae (Qu & Sako, 2023), but we consider a more realistic set of multiple object types. The labeled and unlabeled data are derived from the PLAsTiCC dataset. REDSHIFTS is a new dataset that we contribute as part of this work.

- **Source:** Time-series of bright, nearby labeled objects.

| | AstroClassification | | Redshift | |
|---|---|---|---|---|
| | ID Test Acc (↑) | OOD Acc (↑) | ID Test RMSE (↓) | OOD RMSE (↓) |
| ERM | 71.59±1.10 | 61.26±1.10 | 0.274±0.016 | 0.320±0.009 |
| Standard fine-tuning | 78.84±0.97 | 67.84±0.70 | **0.246±0.015** | 0.277±0.004 |
| ERM + targeted augs | 68.75±0.95 | 67.54±0.32 | 0.310±0.006 | 0.286±0.007 |
| Self-Training | 77.72±0.59 | 65.15±0.67 | 0.304±0.010 | 0.289±0.003 |
| Connect Later | **80.54±1.20** | **79.90±0.60** | 0.256±0.005 | **0.247±0.005** |

Table 2: ID and OOD accuracy (%) for AstroClassification and RMSE for Redshifts of each method. Results are averaged over 5 trials and rows with means within 1 STD of the best mean are bolded.

- **Target:** Time-series of all observed objects from the telescope, often faint and distant (higher redshift).
- **Targeted Augmentation:** Redshifting (example in Section 4).
- **Task:** Redshift regression

**Wildlife Species Classification (iWILDCAM-WILDS).** For iWILDCAM-WILDS (Beery et al., 2020; Sagawa et al., 2022), the task is to identify the wildlife species from static camera trap images. These cameras are placed in a wide variety of environments, which all have unique habitat conditions and camera positions (e.g., African savannah vs. tropical rainforest). In this dataset, we use labeled data from 243 camera traps to learn a model that can generalize to data from 48 unseen camera traps.

- **Source:** 243 camera traps
- **Target:** 48 camera traps from unseen locations
- **Targeted Augmentation:** We augment the labeled dataset with the Copy-Paste Same Y algorithm, which uses image segmentation to copy-paste the animal onto different background images from cameras that have observed the same species (Gao et al., 2023).
- **Task:** 182-class wildlife species classification

**Tumor Detection (CAMELYON17-WILDS).** The task in CAMELYON17-WILDS (Bandi et al., 2018) is to classify whether a patch of a histopathology slide contains a tumor. These slides are contributed from multiple hospitals, which use different stain colors and also vary in distributions of patient cancer stage.

- **Source:** Hospitals 1-3.
- **Target:** Hospitals 4 and 5.
- **Targeted Augmentation:** We augment the labeled dataset with the Stain Color Jitter algorithm, which jitters the color of the slide image in the hematoxylin and eosin staining color space (Tellez et al., 2018).
- **Task:** Binary classification of whether a slide contains a tumor.

# 6 EXPERIMENTS

**Training procedure.** For AstroClassification and Redshifts, we perform pretraining with the masked autoencoding objective, masking 60% of the observations from each light curve. The same pretrained model is used for both tasks to demonstrate the reusability of pretrained features. For iWILDCAM-WILDS, we use a ResNet-50 model pretrained on unlabeled ImageNet data with the SwAV contrastive learning algorithm (Caron et al., 2020). We use a DenseNet121 pretrained on the unlabeled data provided in Sagawa et al. (2022) with SwAV for CAMELYON17-WILDS. We fine-tune the pretrained models with linear probing then fine-tuning (LP-FT) (Kumar et al., 2022), which has been shown to improve OOD performance.

**Baselines.** We evaluate our framework against three baselines: ERM, ERM+targeted augs, and standard fine-tuning. We also include a self-training baseline for AstroClassification and Redshifts, which has been shown to perform well on some real-world datasets (Sagawa et al., 2022). For the self-training baseline, we pseudo-label the target dataset with a trained ERM+targeted augs model, then perform the same targeted augmentation on the pseudo-labeled target dataset. We then train a model with the pseudo-labeled and augmented target dataset combined with the labeled source dataset. We include additional domain adaptation baselines for iWILDCAM-WILDS and CAMELYON17-WILDS: domain-adversarial neural networks (DANN) (Ganin et al., 2016), correlation alignment (CORAL) (Sun et al., 2016), and Noisy Student (Xie et al., 2020b, iWILDCAM-WILDS only).

|  | iWildCam (Macro F1, ↑) | | Camelyon17 (Avg Acc, ↑) | |
|  | ID Test | OOD Test | ID Val | OOD Test |
| --- | --- | --- | --- | --- |
| ERM | 46.4±0.5 | 30.4±0.6 | 89.3±0.9 | 65.2±1.1 |
| Standard fine-tuning | 46.4±0.8 | 31.2±0.6 | 92.3±0.2 | 91.4±0.9 |
| ERM + targeted augs | **51.4±0.6** | 36.1±0.7 | 96.7±0.0 | 90.5±0.4 |
| DANN (Sagawa et al., 2022) | 48.5±3.2 | 31.9±1.6 | 86.1±1.3 | 64.5±1.2 |
| CORAL (Sagawa et al., 2022) | 40.5±1.6 | 27.9±0.5 | 92.3±0.7 | 62.3±1.9 |
| Noisy Student (Sagawa et al., 2022) | 47.5±1.0 | 32.1±0.8 | - | - |
| Connect Later | **51.7±0.8** | **36.9±0.7** | **98.5±0.0** | **94.9±0.4** |

Table 3: ID and OOD performance for each method on IWILDCAM-WILDS and CAMELYON17-WILDS. Results are averaged over 15 trials for IWILDCAM-WILDS and 20 trials for CAMELYON17-WILDS, and we report 95% confidence intervals on each mean estimate. Rows with means within 1 interval of the best mean are bolded.

## 6.1 MAIN RESULTS

Tables 2 and 3 compare the results of Connect Later with baseline methods. We show that Connect Later substantially outperforms all other variants on the OOD metric, including state-of-the-art performances on ASTROCLASSIFICATION by 3% OOD, IWILDCAM-WILDS by 0.8% OOD for ResNet-50, and CAMELYON17-WILDS by 1.1% OOD for DenseNet121.

**AstroClassification.** For ASTROCLASSIFICATION, standard fine-tuning provides a significant performance boost over ERM: $71.6\% \to 78.9\%$ ID, $61.3\% \to 67.8\%$ OOD. ERM+targeted augs underperforms in ID accuracy compared to ERM alone ($71.6\% \to 68.8\%$) and standard fine-tuning ($78.9\% \to 68.8\%$), likely due to relatively strong targeted augmentations toward the redshift distribution of distant, faint objects present in the target distribution. However, OOD accuracy of ERM+targeted augs is competitive with standard fine-tuning, outperforming ERM. Self-training improves both ID and OOD performance compared to ERM but underperforms standard fine-tuning in both domains. Connect Later outperforms the best baseline, standard fine-tuning, by 12% OOD and 2% ID. The ID accuracy improves over standard fine-tuning despite the drop in ID accuracy from adding targeted augmentations to ERM, showing a complementary benefit between pretraining and targeted augmentations. Connect Later improves the state-of-the-art OOD performance on ASTROCLASSIFICATION by 3% over a heavily tuned random forest model with expert-designed features (Boone, 2019).

**Redshifts.** Similarly to ASTROCLASSIFICATION, standard fine-tuning significantly improves over ERM in both ID ($0.27 \to 0.25$, 7% relative) and OOD ($0.32 \to 0.28$, 13% relative) RMSE. Self-training performs similarly to ERM+targeted augs, while Connect Later outperforms the best baseline variant, standard fine-tuning, by 0.03 RMSE (11% relative) with comparable ID error. We use the same pretrained model for both ASTROCLASSIFICATION and REDSHIFTS for standard fine-tuning and Connect Later, demonstrating the reusability of pretrained representations.

**iWildCam.** On IWILDCAM-WILDS, standard fine-tuning does not improve over ERM in either ID or OOD performance, while ERM+targeted augmentations improves by about 6% ID and 6% OOD over ERM and standard fine-tuning. While 2 of the domain adaptation baselines made ID and OOD improvements over standard fine-tuning (DANN: $45.7\% \to 48.5\%$ ID, $30.9\% \to 31.9\%$ OOD; Noisy Student: $45.7\% \to 47.5\%$ ID, $30.9\% \to 32.1\%$ OOD), both fall short of ERM+targeted augs. Connect Later improves over both standard fine-tuning ($30.9\% \to 37.2\%$) and ERM+targeted augmentations ($36.3\% \to 37.2\%$) in OOD performance, achieving a new state-of-the-art performance for ResNet-50 on the IWILDCAM-WILDS benchmark.

**Camelyon17.** On CAMELYON17-WILDS, standard fine-tuning produces significant gains over ERM in both ID ($89.3\% \to 92.3\%$) and OOD ($65.2\% \to 91.4\%$) average accuracy. ERM+targeted augmentations outperforms standard fine-tuning in ID accuracy ($92.3\% \to 96.7\%$), but does not improve OOD. DANN underperforms ERM in both ID and OOD accuracy, while CORAL produces similar ID accuracy as standard fine-tuning but poor OOD performance. Connect Later outperforms the best baseline ID performance by 1.8% (ERM+targeted augs, $96.7\% \to 98.5\%$) and the best OOD performance by 3.5% (standard fine-tuning, $91.4\% \to 94.9\%$). Connect Later also outperforms the current state-of-the-art on DenseNet121, ICON, by 1.1% OOD ($93.8\% \to 94.9\%$).

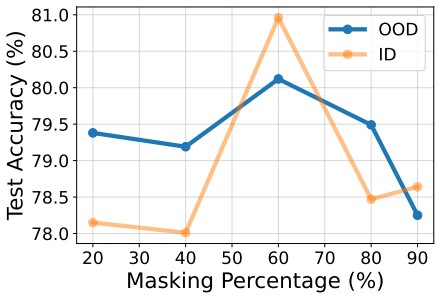

Figure 3: On the ASTROCLASSIFICATION task, Connect Later is relatively robust to pretraining masking percentage both ID and OOD, but 60% masking performs best out of the percentages we tested.

## 6.2 ABLATIONS

We performed ablations on the model size, strength of pretraining augmentations (masking percentage for masked autoencoding), and LP-FT on ASTROCLASSIFICATION. We find that downstream performance is quite robust to masking percentage, while scaling up model size and LP-FT improve performance for pretrained models.

**Model scale.** We tested Connect Later with a larger model ($\sim 3\times$ the parameters of our model, $21M \rightarrow 69M$), and find that scaling up model size improves both ID and OOD accuracy (Table 4). This suggests that scaling up the model is a promising way to further improve performance with Connect Later.

**Strength of pretraining augmentations (masking percentage).** We vary the strength of pretraining augmentations, which changes the connectivity between domains. We tested pretraining masking percentages $\{20, 40, 60, 80, 90\}$% while keeping the masking strategy unchanged (replace 10% of masked indices with random values from the lightcurve, another 10% are kept unchanged, and 80% are replaced with the mask token, which we choose to be 0). We show the ID and OOD test accuracy of each variant in Figure 3. Both ID and OOD performance peak at 60% masking, although we find that the performance of Connect Later is quite robust to the masking percentage, particularly for OOD performance. All of the masking percentages we tried improve on OOD performance over standard fine-tuning or ERM with targeted augmentations. Particularly, even with the strongest pretraining augmentations (90% masking), which should connect the domains more, the OOD performance did not improve over weaker augmentations. We hypothesize that increasing the strength of generic augmentations may indiscriminately increase the connectivity between all source and target examples, including examples from different classes that should not be strongly connected.

**Linear probing then fine-tuning.** Kumar et al. (2022) showed that linear probing (with fixed neural embeddings) and then fine-tuning (LP-FT) the entire model improves both ID and OOD performance. Intuitively, full fine-tuning with a randomly initialized linear probe can destroy the pretrained features, and training the linear probe first mitigates this. We test LP-FT against FT only (all model weights are fine-tuned)

| Number of Parameters | ID Accuracy (↑) | OOD Accuracy (↑) |
|---|---|---|
| 21M (default) | 78.47 | 79.49 |
| 69M | 80.38 | 80.55 |

Table 4: Scaling up model size of Connect Later produces improvements in both ID and OOD performance on the ASTROCLASSIFICATION task.

| | Connect Later | | ERM+targeted augs | |
| | ID Accuracy (↑) | OOD Accuracy (↑) | ID Accuracy (↑) | OOD Accuracy (↑) |
|---|---|---|---|---|
| FT only | 78.07 | 78.6 | 77.88 | 68.43 |
| LP-FT | 78.47 | 79.49 | 65.68 | 67.07 |

Table 5: Linear probing (LP) in addition to fine-tuning (FT) hurts performance for the ERM+targeted augs model but improves performance for Connect Later (tested on the ASTROCLASSIFICATION task).

with the Connect Later model and the ERM+targeted augs baseline. We find that LP-FT improves OOD accuracy by 0.9% over FT only when applied to Connect Later on AstroClassification (Table 5). On the other hand, LP-FT decreased OOD accuracy by 1.4% when applied to ERM+targeted augs, which uses random initialization (no pretraining). As a result, we use LP-FT on pretrained models but not on ERM or ERM+targeted augs.

## 7 Discussion and Related Work

**Augmentations for pretraining.** Data augmentations such as cropping or masking have been vital to semi- and self-supervised learning objectives. Masking or noising the data and training a model to reconstruct the original inputs have been shown to produce useful pretrained representations across multiple modalities (Devlin et al., 2019; Lewis et al., 2020; He et al., 2022; Raffel et al., 2019; Chen et al., 2020; He et al., 2020; Caron et al., 2020). In contrastive learning, models are trained to distinguish augmented "views" of the same input from views of a different input (Chen et al., 2020; Caron et al., 2020; He et al., 2020). Our results demonstrating inconsistent OOD performance across datasets brings up the important future question of how to choose the best pretraining augmentation and algorithm for learning transferable representations.

**Augmentations for robustness.** Data augmentation has been used to improve model robustness and avoid catastrophic failures due to spurious, label-independent changes (e.g. translation or rotation in vision) (Hendrycks et al., 2019; Rebuffi et al., 2021; Ng et al., 2020). The augmentation strategies used in prior work are generic perturbations that aim to increase the diversity of inputs (e.g., Simard et al., 2003; Krizhevsky et al., 2012; Cubuk et al., 2019; 2020; DeVries & Taylor, 2017; Zhang et al., 2017), though a number of studies have shown that the type of data augmentations matters for performance (Chen et al., 2020; Xie et al., 2020a). Augmentations have also been leveraged in the self-training paradigm, which improves generalization to unseen data by training on the pseudo-labeled full dataset (Xie et al., 2020b; Sohn et al., 2020; Yang et al., 2021). We show that a self-training baseline with pseudo-labels from an ERM+targeted augs model does not outperform Connect Later, indicating that pretraining is an important component of the framework. Connect Later exposes targeted augmentations as a design interface for improving robustness with knowledge of the distribution shift, leveraging pretrained representations.

**Targeted augmentations.** In problems with domain shift, Gao et al. (2023) show that targeted augmentations outperform generic augmentations on unseen data. They identify spurious domain-dependent, label-independent features in the source dataset and construct targeted augmentations by randomizing these features. Gao et al. (2023) consider the domain generalization setting, in which no data from the target dataset is available. We consider targeted augmentations in the domain adaptation setting, in which we can model the target distribution of these spurious features with the unlabeled target data. In general, designing targeted augmentations specific to each distribution shift may be difficult and require expert guidance. As part of the Connect Later framework, we provide a general methodology for the design of such augmentations. Certain aspects, such as the selection of feature space $z$ and transformation distribution $T$ could be learned from the unlabeled data itself, which we leave for future work. We also show that targeted augmentations better leverage pretrained representations for complementary gains in OOD performance.

## 8 Conclusion

We show that pretraining with generic augmentations is not a panacea for all distribution shifts and tasks, and sometimes does not outperform supervised learning on labeled source data. Pure supervised learning, however, does not use the unlabeled data or produce reusable representations. The Connect Later framework allows for better leverage of pretrained representations for OOD performance by applying targeted augmentations at fine-tuning time. Future work could focus on learning targeted augmentations from the source and target data distributions as well as further understanding of how the choice of pretraining augmentations affects downstream ID/OOD performance.

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

## A  ADDITIONAL DATASET DETAILS

### A.1  ASTROCLASSIFICATION, REDSHIFTS DATASETS

The ASTROCLASSIFICATION and REDSHIFTS datasets were adapted from the 2019 Photometric LSST Astronomical Time-Series Classification Challenge (The PLAsTiCC team et al., 2018) [1]. This diverse dataset contains 14 types of astronomical time-varying objects, simulated using the expected instrument characteristics and survey strategy of the upcoming Legacy Survey of Space and Time (LSST Ivezić et al., 2019) conducted at the Vera C. Rubin Observatory. It includes two overall categories of time-series objects: *transients*, short-lived events such as supernovae, and *variable* sources, those with fluctuating brightness such as pulsating stars. Specifically, the dataset includes the following transients: type Ia supernovae (SNIa), SNIax, SNIa-91bg, SNIbc, SNII, superluminous supernovae (SLSN), tidal disruption events (TDE), and single lens microlensing events ($\mu$Lens-Single); and the following variable objects: active galactic nuclei (AGN), Mira variables, eclipsing binary systems (EB), and RR Lyrae (RRL).

Millions of potential new objects are discovered per observing night, and important metadata such as object type, redshift, or other physical parameters, require astronomers to take time-intensive *spectra* of each object. Spectra are a granular brightness vs. wavelength measurement at a single point in time, and are typically only taken for bright, nearby objects which require less exposure time than faint, faraway objects. The vast majority of discovered objects will only have a time series of imaging data taken in 6 broad

---

[1]https://zenodo.org/record/2539456

wavelength ranges, or *photometric bands*. The time variation of these objects in these coarse wavelength bands does offer important clues about these physical parameters, but the labeled training data for both ASTROCLASSIFICATION and REDSHIFTS come from the unrepresentative subset of objects with spectra.

In these tasks, we are specifically interested in predicting the object type (e.g. type II supernova) and the cosmological redshift of these objects in the unlabeled dataset. *Cosmological redshift* is a proxy for distance in the universe, and an important piece of metadata for understanding an object's physical processes as well as other applications, such as estimating the expansion rate of the universe with type Ia supernovae.

**Problem Setting.** The task is to predict object type for ASTROCLASSIFICATION (redshift for REDSHIFTS) from time-series of object brightness. The input $x$ consists of flux measurements and associated uncertainties at times $\boldsymbol{t}$ and photometric band that each measurement was taken in $\boldsymbol{b}$: $\{F(t_i, b_j)\}_{i=1, j=1}^{T,W}, \{F_{\text{err}}(t_i, b_j)\}_{i=1, j=1}^{T,W}$. For this work, we map each $b \in \boldsymbol{b}$ to the central wavelength of the $b$ band, which we denote $\boldsymbol{w}$. The domain $d$ is binary, corresponding to whether the object has a spectrum (and thus a label). The labels $y$ are available only for objects with spectra, and are one of 14 types of astronomical time-varying objects for ASTROCLASSIFICATION (redshift of the object for REDSHIFTS). We seek to optimize performance on the unlabeled data, which are generally fainter and further away than the labeled subset. We evaluate on these examples as well as held-out examples from the labeled subset.

**Data.** The training set of 7,846 objects is designed to emulate a sample of objects with spectra and thus biased toward brighter, more nearby objects compared to the test set of 3,492,888 objects. A random subset of 10,000 test set objects was selected for evaluation.

1. **Source:** 6,274 objects
2. **ID Test**: 782 objects
3. **OOD Test:** 10,000 objects

All data were simulated with the SuperNova ANAlysis (SNANA, Kessler et al., 2009) software library. Further details about the astrophysical models and LSST instrument characteristics used in the simulation can be found in Kessler et al. (2019).

# B DATA AUGMENTATIONS

## B.1 GENERIC AUGMENTATIONS FOR PRETRAINING

**AstroClassification and Redshifts.** For the ASTROCLASSIFICATION and REDSHIFTS datasets, we randomly mask a subset of the input sequence using the masked language modeling paradigm introduced by Devlin et al. (2019). Given an unlabeled input sequence $x$, a training input $x'$ can be generated by randomly masking elements of $x$ while the associated label $y$ consists of the original, unmasked values. The model is trained to use contextual information (unmasked elements) to successfully reconstruct most of the sequence. From our ablation experiments, we find that a masking percentage of 60% produces the best downstream results. We follow an existing implementation for astronomical time-series (Donoso-Oliva et al., 2023) and set 80% of the masked elements to 0, replace 10% with a random element from the sequence, and keep the remaining 10% unchanged.

**iWildCam and Camelyon17.** We use a ResNet-50 model for iWildCam pretrained on ImageNet with SwAV, a contrastive learning algorithm Caron et al. (2020). For Camelyon17, we use a DenseNet121 pretrained with SwAV on the unlabeled CAMELYON17-WILDS dataset from Sagawa et al. (2022). SwAV uses random cropping augmentations of different resolutions.

## B.2 TARGETED AUGMENTATIONS FOR FINE-TUNING

**Redshifting for AstroClassification and Redshifts.** The OOD test set of the ASTROCLASSIFICATION and REDSHIFTS datasets have many more high redshift objects than the source dataset, leading us to adopt an augmentation scheme to alleviate this shift. Figure 4 shows the redshift distributions of the source, augmented, and target datasets. Redshifting places each object at a new redshift and recomputes its light curve sampling, fluxes, and flux uncertainties accordingly. This augmentation algorithm was adapted from Boone (2019).

An input $\boldsymbol{X} \in \mathbb{R}^{T \times W}$ is a multivariate time series of flux values at specified times and observed wavelengths, $\{F(t_i, w_j)\}_{i=1, j=1}^{T,W}$. We also have $\boldsymbol{X}_{\text{err}} \in \mathbb{R}^{T \times W}$, representing the flux errors corresponding to each element of $\boldsymbol{X}$. We denote the elements of $\boldsymbol{X}'_{\text{err}}$ by $\{F_{\text{err}}(t_i, w_j)\}_{i=1, j=1}^{T,W}$. Our goal is to model $F, F_{\text{err}} : \mathbb{R} \times \mathbb{R} \to \mathbb{R}$ at a new chosen redshift, $z'$, to produce augmented inputs $\boldsymbol{X}', \boldsymbol{X}'_{\text{err}}$.

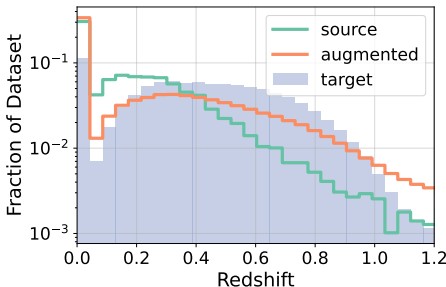

Figure 4: Redshift distributions of source, augmented, and target datasets for the ASTROCLASSIFICATION and REDSHIFTS tasks.

- We first construct a distribution from which to sample the new redshift, taking into account the current redshift of the object $z_{\text{orig}}$ as well as the target redshift distribution. We then sample a new redshift, $z' \sim \text{loguniform}(0.95z_{\text{orig}}, \min(1.5(1+z_{\text{orig}})-1, 5z_{\text{orig}}))$.
- We fit a Gaussian process (GP) model for $F$ with training observations $\boldsymbol{X}$ queried at the training input values $(\boldsymbol{t}, \boldsymbol{w})$, and denote the predictive mean and variance of the GP as $F', F'_{\text{err}}$.
- Given the new redshift value $z'$, we rescale the timestamps and wavelengths of the original observations to account for the physical effects of the new redshift value: $\boldsymbol{t}_{\text{new}} = \frac{1+z'}{1+z_{\text{orig}}}\boldsymbol{t}$, $\boldsymbol{w}_{\text{new}} = \frac{1+z'}{1+z_{\text{orig}}}\boldsymbol{w}$. We also randomly drop out 10% as well as a large swath of $(\boldsymbol{t}_{\text{new}}, \boldsymbol{w}_{\text{new}})$ to simulate distinct observing seasons (telescope observing only occurs in the winter).
- We obtain GP predictions at test inputs $\{F'(t_{\text{new},i}, w_{\text{new},j})\}_{i=1,j=1}^{T,W}$, $\{F'_{\text{err}}(t_{\text{new},i}, w_{\text{new},i})\}_{i=1,j=1}^{T,W}$ and scale them by the log ratio of the new and original distances:
$$\tilde{\boldsymbol{X}}' = 10^{0.4(d(z')-d(z_{\text{orig}}))}\{F'(t_{\text{new},i}, w_{\text{new},j})\}_{i=1,j=1}^{T,W},$$
$$\tilde{\boldsymbol{X}}'_{\text{err}} = 10^{0.4(d(z')-d(z_{\text{orig}}))}\{F'_{\text{err}}(t_{\text{new},i}, w_{\text{new},j})\}_{i=1,j=1}^{T,W},$$
where $d(z)$ is the distance corresponding to redshift $z$.
- We roughly model the observational noise of the telescope from the target data as a function of wavelength and sample $\epsilon \in \mathbb{R}^W$ from it. We define
$$\boldsymbol{X}' = \{\tilde{\boldsymbol{X}}'_{:,j} + \epsilon_j\}_{j=1}^{W}, \boldsymbol{X}'_{\text{err}} = \left\{\sqrt{\tilde{\boldsymbol{X}}'^2_{\text{err},:,j} + \epsilon_j^2}\right\}_{j=1}^{W}.$$
- We model the observational capabilities of the telescope to ensure that our augmented input $\boldsymbol{X}', \boldsymbol{X}'_{\text{err}}$ does not fall below the threshold of detection. We "accept" an augmented input $\boldsymbol{X}', \boldsymbol{X}'_{\text{err}}$ if the signal-to-noise ratio (SNR) of at least two observations is over 5, i.e. $\text{SNR}(\boldsymbol{X}'_{i,j}, \boldsymbol{X}'_{\text{err},i,j}) \geq 5$ for at least 2 of $i \in \{1,...,T\}, j \in \{1,...,W\}$. We define $\text{SNR}(x, x_{\text{err}}) = \frac{|x|}{x_{\text{err}}}$.

**Copy-Paste (Same Y) for iWildCam.** This augmentation strategy randomizes the backgrounds of wildlife images to reduce the model's dependence on these spurious features for species classification. Specifically, a segmentation mask is applied to each image to separate the animal from the background, and the animal is "copy-pasted" into a new background from a camera that has observed that animal species. This was the best performing augmentation strategy from Gao et al. (2023).

**Stain Color Jitter for Camelyon17.** This augmentation, originally from Tellez et al. (2018), alters the pixel values of the slide images to emulate different staining procedures used by different hospitals. The augmentation uses a pre-specified Optical Density (OD) matrix to project images from RGB space to a three-channel hematoxylin, eosin, and DAB space before applying a random linear combination. This was the best performing augmentation strategy from Gao et al. (2023).

## C   EXPERIMENTAL DETAILS

**AstroClassification and Redshifts.** We use an encoder-only Informer model (Zhou et al., 2021) with 8 encoder layers of 12 attention heads each. The model hidden dimension was chosen to be 768 and the layer MLPs have hidden dimension 256. Due to the 2-dimensional position data (each element of the time-series has an associated time and photometric band/wavelength) and irregular sampling of our dataset, we train

a positional encoding based on learnable Fourier features following Li et al. (2021). We also select a random window of length 300 from each example (and zero-pad examples with fewer than 300 observations) to produce inputs of uniform shape. We perform pretraining with a batch size of 256 and learning rate 1e-4 (selected from 1e-3 ∼ 1e-6) for 75,000 steps. We finetune the pretrained model with linear probing for 20,000 steps (for pretrained models only) and learning rate 1e-4, then fine-tuning for 10,000 steps at learning rate of 4e-5. We increase the learning rate for models without pretraining to 1e-4 for FT. The REDSHIFTS task uses LP learning rate of 5e-4 and FT learning rate of 1e-4. We decrease the learning rate per step with a linear scheduler.

**iWildCam.** For pretraining, we use ResNet-50 pretrained on ImageNet with SwAV (Caron et al., 2020). During fine-tuning, we train all models for 15 epochs with early stopping on OOD validation performance, following Gao et al. (2023). For pretrained models, we also do 10 epochs of linear probing before fine-tuning (LP-FT, Kumar et al., 2022) for 15 epochs, where the linear probe is trained with Adam and the linear probe weights used to initialize the fine-tuning stage is chosen with OOD validation performance. To reduce the noise in OOD results, for all methods we select the epoch in the last 5 epochs with the best OOD validation performance and report OOD test results with that version of the model. Following Gao et al. (2023), we allow for 10 hyperparameter tuning runs, where we sample the following hyperparameters independently from the following distributions: the linear probe learning rate ($10^{\text{Uniform}[-3,-2]}$), fine-tuning learning rate ($10^{\text{Uniform}[-5,-2]}$), and probability of applying the augmentation (Uniform$[0.5,0.9]$) and pick the hyperparameter configuration with the best OOD validation performance. For ERM and ERM+targeted augmentations, we use the tuned hyperparameters from Gao et al. (2023). To decrease the confidence interval, all reported performances for ERM, ERM+targeted augs, standard fine-tuning, and Connect Later are averaged over 15 seeds. DANN, CORAL, and Noisy Student results are averaged over 5 seeds.

**Camelyon17.** For pretraining, we use DenseNet121 pretrained on the unlabeled CAMELYON17-WILDS dataset presented in Sagawa et al. (2022) with SwAV (Caron et al., 2020). During fine-tuning, we train all models for 15 epochs with early stopping on OOD validation performance, following Gao et al. (2023). For pretrained models, we also do 10 epochs of linear probing before fine-tuning (LP-FT, Kumar et al., 2022) for 15 epochs, where the linear probe is trained with Adam and the linear probe weights used to initialize the fine-tuning stage is chosen with OOD validation performance. To reduce the noise in OOD results, for all methods we select the epoch with the best OOD validation performance and report OOD test results with that version of the model. Following Gao et al. (2023), we allow for 10 hyperparameter tuning runs, where we sample the following hyperparameters independently from the following distributions: the linear probe learning rate ($10^{\text{Uniform}[-3,-2]}$), fine-tuning learning rate ($10^{\text{Uniform}[-5,-2]}$), probability of applying the augmentation (Uniform$[0.5,0.9]$), and augmentation strength (Uniform$[0.05,0.1]$), and pick the hyperparameter configuration with the best OOD validation performance. All results are averaged over 20 seeds.

## D SIMPLE CONSTRUCTION WHERE CONNECT LATER IMPROVES OVER PRETRAINING OR TARGETED AUGMENTATIONS ALONE.

We give a simple construction for constrastive pretraining based on the construction in Proposition 3 (Appendix A.2) of Shen et al. (2022), where Connect Later improves over pretraining (standard fine-tuning) or targeted augmentations alone.

**Data distribution.** We consider binary classification with 2 domains. Let $\mathcal{S} = \{x \in \mathcal{X} : d_x = 1\}$ and $\mathcal{T} = \{x \in \mathcal{T} : d_x = 2\}$, and assume that $P_S$ and $P_T$ are uniform over $\mathcal{S}$ and $\mathcal{T}$. The unlabeled distribution for pretraining is the uniform distribution over $\mathcal{X}$. The source domain $\mathcal{S} = \{1,2\}$ contains 2 points and the target domain $\mathcal{T} = \{3,4,5,6,7,8\}$ contains 6 points. For simplicity, we let the labels $y_x$ be a deterministic function of the input $x$. The label space is $\mathcal{Y} = \{-1,1\}$. The label for $x \in \{1,3,5,7\}$ is $y_x = 1$ and the label for $x \in \{2,4,6,8\}$ is $y_x = -1$. Only the source data is labeled.

**ERM with targeted augmentations.** ERM with targeted augmentations applies the fine-tuning objective (Equation 2) without prior pretraining on unlabeled data. To specialize to this section, the ERM objective is

$$\mathcal{L}_{\text{ERM}}(f) = \mathbb{E}_{x \sim P_S, x' \sim \mathcal{A}_{\text{ft}}(\cdot|x)}[\ell(f(x'),y_x)]. \tag{3}$$

ERM returns a classifier $\widehat{f}_{\text{erm}} \in \arg\min_f \mathcal{L}_{\text{ERM}}(f)$.

**Spectral contrastive learning.** Following HaoChen et al. (2021) and Shen et al. (2022), we analyze contrastive learning from an augmentation graph perspective, where inputs $x$ are connected via augmentations with edge weights $S_+(x,x')$, which represent the probability of $x,x'$ being a positive pair (augmentations

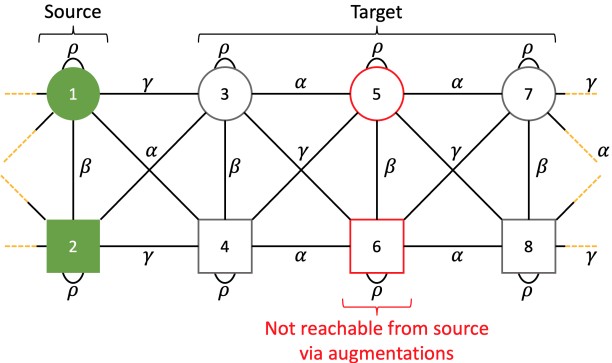

Figure 5: Example distribution of data and augmentations for contrastive learning where Connect Later improves OOD performance over contrastive pretraining+standard fine-tuning and ERM+targeted augmentations. The augmentation graph is similar to Shen et al. (2022) except the edge weights connecting 1,2 and 3,4 are swapped. The shapes represent classes, while the labeled data is shaded in green. The generic augmentation probabilities are marked as edge weights, where we assume that $\alpha > \gamma + \beta$. Here, targeted augmentations which first swap inputs 1 and 2 before applying a generic augmentation help to align the source and target. However, some target inputs are not reachable via augmentations from source inputs. Standard fine-tuning can generalize throughout the target domain, but only in conjunction with targeted augmentations that align the source and target. The orange dotted lines on the far ends connect to each other (the graph wraps around).

of the same input $x$). For theoretical analysis, we analyze the spectral contrastive learning objective:

$$\mathcal{L}_{\text{pretrain}}(\phi) = -2 \cdot \mathbb{E}_{(x,x^+) \sim S_+} \left[ \phi(x)^\top \phi(x^+) \right] + \mathbb{E}_{x,x' \sim P_U} \left[ \left( \phi(x)^\top \phi(x') \right)^2 \right]. \tag{4}$$

The result of pretraining to optimize the above objective is an encoder $\widehat{\phi} : \mathcal{X} \to \mathbb{R}^k$.

**Linear probing (fine-tuning step).**    Instead of analyzing fine-tuning, we follow Shen et al. (2022) and analyze linear probing on top of the pretrained representations from the encoder. We train a linear model with parameters $B \in \mathbb{R}^{r \times k}$, where $r$ is the number of classes. We minimize the objective:

$$\mathcal{L}(B) = \mathbb{E}_{x \sim P_S} \left[ \ell(B \widehat{\phi}(x), y_x) \right] + \eta \|B\|_F^2, \tag{5}$$

where $\ell$ is the squared loss and we take $y_x \in \mathbb{R}^k$ to be a one-hot encoding of the class label. The resulting classifier is $\widehat{f}(x) = \text{argmax}_{i \in [r]} (\widehat{B} \widehat{\phi}(x))_i$.

**Pretraining augmentations (Figure 5)**    We define the pretraining augmentation distribution $\mathcal{A}_{\text{pre}}(\cdot \,|\, x)$ to be

$$\mathcal{A}_{\text{pre}}(x' \,|\, x) = \begin{cases} \rho' & x = x' \\ \alpha' & \{x', x\} \in \{\{1,4\}, \{3,5\}, \{5,7\}, \{2,5\}, \{4,6\}, \{6,8\}, \{1,8\}, \{2,7\}\} \\ \beta' & \{x', x\} \in \{\{1,2\}, \{3,4\}, \{5,6\}, \{7,8\}\} \\ \gamma' & \{x', x\} \in \{\{1,3\}, \{2,4\}, \{3,6\}, \{4,5\}, \{5,8\}, \{6,7\}, \{1,7\}, \{2,8\}\} \end{cases} . \tag{6}$$

Notice that the weight between 1,3 is $\gamma'$ and the weight between 1,4 is $\alpha'$, and the weights are similarly swapped for 2,4, and 2,5. We assume that $\rho', \alpha', \beta',$ and $\gamma'$ are in $(0,1)$ and are distinct. We also assume that the augmentation probabilities satisfy $\rho' > \max\{\alpha', \beta'\}$ and $\min\{\alpha', \beta'\} > \gamma'$. Following Shen et al. (2022), we can convert these to positive pair probabilities $\rho, \alpha, \beta, \gamma$ with similar properties by renormalizing.

Given the above setting, the following is a simplified form of Proposition 3 from Shen et al. (2022), if we instead use the following augmentation distribution, which swaps the edge weight magnitudes that involve nodes 1 and 2:

$$\mathcal{A}_{\text{prop}}(x' \,|\, x) = \begin{cases} \rho' & x = x' \\ \alpha' & \{x', x\} \in \{\{1,3\}, \{3,5\}, \{5,7\}, \{2,4\}, \{4,6\}, \{6,8\}, \{1,7\}, \{2,8\}\} \\ \beta' & \{x', x\} \in \{\{1,2\}, \{3,4\}, \{5,6\}, \{7,8\}\} \\ \gamma' & \{x', x\} \in \{\{1,4\}, \{2,3\}, \{3,6\}, \{4,5\}, \{5,8\}, \{6,7\}, \{1,8\}, \{2,7\}\} \end{cases} . \tag{7}$$

**Proposition 1** (Shen et al. (2022)). *With the above construction for the input space $\mathcal{X}$, unlabeled distribution $P_U$, and data augmentation $\mathcal{A}_{prop}$, for some feature dimension $k \in \mathbb{Z}^+$ a linear probe trained on contrastive pre-*

*trained features achieves 0 target error: $\mathcal{L}_{0-1}(\widehat{f})=0$. However, for all $k\in\mathbb{Z}^+$, there exists a minimizer $\widehat{f}_{erm}$ of the ERM objective (with data augmentations according to $\mathcal{A}_{prop}$) that has non-zero error: $\mathcal{L}_{0-1}(\widehat{f}_{erm})=1/3$.*

**ERM with targeted augmentations can get high OOD error.** In general, we proceed by defining the following targeted augmentation, which allows us to reduce to the setting of Proposition 1:

$$\mathcal{A}_{\text{ft}}(x'\,|\,x)=\begin{cases} 1 & \{x',x\}\in\{1,4\},\{2,3\} \\ 1 & x=x' \text{ and } x\notin\{1,2\} \\ 0 & \text{otherwise} \end{cases} \tag{8}$$

which transforms input 1 to 4 and the input 2 to 3, while keeping all other inputs the same. Since the ERM with augmentations objective will not contain a term involving inputs 5,6,7, or 8 and thus the prediction on these inputs do not affect the objective, there exists a minimizer of the ERM objective (Equation 3) that predicts the wrong label for inputs 5,6,7,8 and has target error 2/3. This is because these nodes are unreachable via augmentations of the source inputs, and thus the ERM objective can be minimized with any arbitrary prediction on these inputs.

**Standard fine-tuning has high OOD error.** By Proposition 1, standard fine-tuning after contrastive pretraining has zero target (OOD) error when the pretraining augmentations do not have swapped edges. By symmetry, standard fine-tuning (contrastive pretraining + linear probing) on our augmentation graph with pretraining augmentations $\mathcal{A}_{\text{pre}}$ outputs the opposite label for all target inputs, resulting in an OOD error of 1. This is because the source and target domains are misaligned in our augmentation graph.

**Connect Later achieves zero OOD error.** Connect Later applies targeted augmentations $\mathcal{A}_{\text{ft}}$ during the linear probing step (on top of contrastive pretrained representations). This choice of targeted augmentations reduces to the setting of Proposition 1 where the labeled source domain consists of the inputs 3,4 instead. By the symmetry of the graph and applying Proposition 1, Connect Later achieves 0 OOD error.

