# OpenReview forum: "Connect Later: Improving Fine-Tuning for Robustness with Targeted Augmentations"
_ICLR.cc/2024/Conference — Submitted to ICLR 2024_

### Official Review · Reviewer_erbX · 2023-10-30

**Soundness:** 3 good
**Presentation:** 3 good
**Contribution:** 2 fair
**Rating:** 5
**Confidence:** 4

**Summary:**

The paper proposes Connect Later to improve model robustness in domain adaptation scenarios. The approach first leverages generic augmentations to pretrain on combined unlabeled source and target data. It then employs carefully designed targeted augmentations during fine-tuning on labeled source data to better connect the source and target domains based on knowledge of their distribution shift. The experiment shows the effectiveness of Connect Later over several baselines on three datasets.

**Strengths:**

The paper is clearly written in general, so the overall quality of the paper is satisfactory.

**Weaknesses:**

1. The proposed framework hinges on the manual design of the transformation distribution and requires knowledge about the distribution shift between source and target domains to create effective targeted augmentations. This property limits the applicability of the proposed method in scenarios where the shift is unknown or diverse.

2. On iWildCam, Connect Later only marginally outperforms ERM+targeted augmentations, which indicates that if there is diverse distribution shifts, the proposed method is not quite effective.

**Questions:**

Could the authors test the sensitivity of the performance of Connect Later with respect to the design choice of transformation distribution?

---

> ### Author Response · Authors · 2023-11-19
>
> We thank the reviewer for the feedback. erbX notes that “the paper is clearly written” and “the experiment shows the effectiveness of Connect Later over several baselines on three datasets”. We address specific points below:
>
> > On iWildCam, Connect Later only marginally outperforms ERM+targeted augmentations, which indicates that if there is diverse distribution shifts, the proposed method is not quite effective.
>
> We ran additional seeds for iWildCam to clarify these results, for a total of 15 seeds. We report 95% confidence intervals on the mean estimate in parentheses and demonstrate that **the mean F1 results of ERM+targeted augs are outside of the 95% confidence intervals of Connect Later.**
>
> |                      | ID Test Macro F1 | OOD Test Macro F1 |
> |---|---:|---:|
> | ERM                  | 46.4 (0.5)       | 30.4 (0.6)        |
> | Standard fine-tuning | 46.4 (0.8)       | 31.2 (0.6)        |
> | ERM + targeted augs  | 51.4 (0.6)       | 36.1 (0.7)        |
> | DANN                 | 48.5 (3.2)       | 31.9 (1.6)        |
> | CORAL                | 40.5 (1.6)       | 27.9 (0.5)        |
> | Noisy Student        | 47.5 (1.0)       | 32.1 (0.8)        |
> | Connect Later        | 51.7 (0.8)       | 36.9 (0.7)        |
>
> **To further strengthen our results, we tested on another benchmark dataset, Camelyon17-WILDS, and show that Connect Later sets a new state-of-the-art on this benchmark for DenseNet121.** Connect Later applied with the stain color jitter targeted augmentation achieved 94.9% +/- 0.4% accuracy OOD compared to the current state-of-the-art (93.8 +/- 0.2%, cite ICON). **Overall, Connect Later achieves the state-of-the-art OOD performance on iWildCam-WILDS (for ResNet-50), Camelyon17-WILDS (for DenseNet-121), and AstroClassification.**
>
>
> |                      | ID Val Avg Acc | OOD Test Avg Acc |
> |---|---:|---:|
> | ERM                  | 89.3 (0.9)     | 65.2 (1.1)       |
> | Standard fine-tuning | 92.3 (0.2)     | 91.4 (0.9)       |
> | ERM + targeted augs  | 96.7 (0.0)     | 90.5 (0.4)       |
> | DANN                 | 86.1 (1.3)     | 64.5 (1.2)       |
> | CORAL                | 92.3 (0.7)     | 62.3 (1.9)       |
> | ICON (SOTA)          | 90.1 (0.3)     | 93.8 (0.2)       |
> | Connect Later        | **98.5 (0.0)**    | **94.9 (0.4)**       |
>
>
> > Could the authors test the sensitivity of the performance of Connect Later with respect to the design choice of transformation distribution?
>
> - **During the rebuttal period, we verified that Connect Later (with targeted augmentations) outperforms Connect Later with generic augmentations at fine-tuning time in OOD performance, tested on AstroClassification (79.9% vs. 72.7%)**. The difference in ID accuracy between Connect Later and standard fine-tuning+generic augmentations is much smaller (80.54% vs 79.9%) in comparison. This suggests that the choice of augmentation is important, especially for OOD performance.
> - For variations on targeted augmentations, Gao et al. 2023 tested a variant (Copy-Paste All Backgrounds) of the targeted augmentation we use for iWildCam (Copy-Paste Same Y). Copy-Paste segments out the animal from the image and pastes it onto a different background. Copy-Paste Same Y requires that the background is from a camera trap that has observed this animal species before, preventing, e.g., camels being pasted in water. Copy-Paste All Backgrounds does not leverage this requirement. **Gao et al. found a 2% degradation in OOD performance with Copy-Paste All Backgrounds compared to Copy-Paste Same Y, indicating that the realism of the targeted augmentation is important.**
> - For pretraining augmentations, we performed an ablation on the masking percentage for the AstroClassification task (Fig 3) and demonstrated that both ID and OOD performance are relatively robust to the chosen masking percentage between 20-90%.
>
> > The proposed framework hinges on the manual design of the transformation distribution and requires knowledge about the distribution shift between source and target domains to create effective targeted augmentations. This property limits the applicability of the proposed method in scenarios where the shift is unknown or diverse.
>
> **We describe a general procedure for designing targeted augmentations from the source and target data distributions in Section 4 of the paper**, as well as an example of designing the redshifting augmentation used for AstroClassification within this framework. While these augmentations do require some manual design, **a relatively small amount of manual design leads to strong empirical results (SOTA on AstroClassification, iWildCam-WILDS, Camelyon17-WILDS + best performance on new dataset Redshifts).**

---

> > ### Author Response · Authors · 2023-11-21
> >
> > Hello!
> >
> > We were wondering if you've had a chance to look over our comments? We experimented with the fine-tuning augmentation design and found that generic augmentations are not able to deliver the OOD performance of targeted augmentations (Connect Later). We also show that Connect Later sets a state-of-the-art on a new benchmark dataset, Camelyon17-WILDS, with DenseNet-121. There are also other experiments and details in the comments above!
> >
> > Please let us know if you have other questions or concerns. Thank you!

---

### Official Review · Reviewer_atYU · 2023-10-31

**Soundness:** 2 fair
**Presentation:** 2 fair
**Contribution:** 2 fair
**Rating:** 5
**Confidence:** 4

**Summary:**

The paper proposes Connect Later to improve robustness. It involves performing self-supervised pretraining (e.g., masked autoencoding or contrastive learning) followed by finetuning with augmentations designed with knowledge of the distribution shift. Supportive results were shown on several real-world dataset.

**Strengths:**

Evaluations were done on real world datasets with supportive results.

**Weaknesses:**

1. Generalizability of the method.
    - If target augmentations involve knowledge of the distribution shift, why pre-training is needed? If we are able to generate the target data, where other options include using powerful generative models [1], why do we need pre-training? It seems to be more a question of when is pre-training necessary.
    - It is also not clear if pre-training in general is useful as the experimental results were only shown with one type of pre-training method and a different pre-training for different datasets. How sensitive are the results to the choice of pre-training method?
    - From Table 2, it seems like the results are sensitive to the target augmentations. For iwilds most of the gains seem to come from the targeted augmentations. But for the astro datasets the ID test acc for astroclassification is better without targeted augmentations. One a new task or dataset, how should one choose the targeted augmentations?
2. Test-time methods, like test-time augmentation/training also makes use of unlabelled data. These methods aim to adapt to the distribution shift as they occur. Furthermore, some of these pre-training objectives like masked auto-encoding have been used for test-time adaptation [2]. It may be beneficial to pose these methods wrt Connect Later.
4. Presentation
    - Sec 4 gives the technical details for creating augmentations before explaining the tasks. It may not be clear what the tasks for Astroclassification and Redshift are about and so the technical details are not easy to understand. I would suggest moving Sec 5 before 4.
    - In Fig 2, top row, what do the colors mean? Why does augmentation introduce errors?
    - It may be useful to have additional columns in the main table of results (e.g. Tab 2), that describes what the different methods are. E.g., whether finetuning augmentations are used, and with checks in the rows for the relevant methods.




----
[1] Diversify Your Vision Datasets with Automatic Diffusion-based Augmentation. NeurIPS’23

[2] Y. Gandelsman, et al. Test-Time Training with Masked Autoencoders. NeurIPS’23

**Questions:**

See the points raised in weaknesses for questions and suggestions.

---

> ### Author Response · Authors · 2023-11-19
> **Response (1/2)**
>
> We thank the reviewer for the feedback. atYU notes that **supportive results were shown on several real-world datasets**. We answer specific questions below:
>
> > If target augmentations involve knowledge of the distribution shift, why pre-training is needed? If we are able to generate the target data, where other options include using powerful generative models [1], why do we need pre-training? It seems to be more a question of when is pre-training necessary.
>
> **The ablations in the paper show that pretraining is necessary**: specifically, ERM+targeted augs removes the pretraining step from Connect Later and results in worse OOD accuracy on all datasets compared to Connect Later (67.5% vs. 79.9% OOD accuracy on AstroClassification, 36.3% vs. 37.2% OOD macro F1 on iWildCam).
> Note that even though targeted augmentations can transform source data to more closely resemble target data, they are not perfect and do not describe the target distribution fully. Pretraining incorporates unlabeled data from the target to learn a better representation of the target data. Generative models such as Stable Diffusion could also be used to more easily design a targeted augmentation, and we view them as complementary to our framework.
>
> > the experimental results were only shown with one type of pre-training method and a different pre-training for different datasets. How sensitive are the results to the choice of pre-training method?
>
> - **Connect Later improves OOD accuracy with both masked autoencoding (AstroClassification, Redshifts) and contrastive pretraining (iWildCam-WILDS, Camelyon17-WILDS).**
> - During the rebuttal period, we ran the standard fine-tuning baseline with SWaV contrastive pretraining on another benchmark, Camelyon17-WILDS, and found that **the choice of pretraining method is not the source of inconsistent OOD results from pretraining.** On Camelyon17-WILDS, contrastive pretraining with standard fine-tuning improves accuracy over both ERM+targeted augs and ERM. This is very different from iWildCam-WILDS, where contrastive pretraining with standard fine-tuning did not even improve accuracy over ERM.
>
> - On Camelyon17-WILDS, Connect Later also achieves the new state-of-the-art ID and OOD accuracy. **Overall, Connect Later achieves the state-of-the-art OOD accuracy on iWildCam-WILDS (for ResNet-50), Camelyon17-WILDS (for DenseNet-121), and AstroClassification.**
>
> |                      | ID Val Avg Acc | OOD Test Avg Acc |
> |---|---:|---:|
> | ERM                  | 89.3 (0.9)     | 65.2 (1.1)       |
> | Standard fine-tuning | 92.3 (0.2)     | 91.4 (0.9)       |
> | ERM + targeted augs  | 96.7 (0.0)     | 90.5 (0.4)       |
> | DANN                 | 86.1 (1.3)     | 64.5 (1.2)       |
> | CORAL                | 92.3 (0.7)     | 62.3 (1.9)       |
> | ICON (previous SoTA)          | 90.1 (0.3)     | 93.8 (0.2)       |
> | Connect Later        | **98.5 (0.0)**     | **94.9 (0.4)**       |
>
> - **Instead, we explain the inconsistency in OOD results of pretraining by differences in the connectivity of domains and classes in each dataset.** During the rebuttal period, we investigated why pretraining was much more effective for OOD performance for AstroClassification than for iWildCam by evaluating connectivity following Shen et al., 2022 [1]. We found that one of the assumptions made in [1] that governs when contrastive pretraining works for UDA was violated for iWildCam, but not for AstroClassification.
>
> - Specifically, we evaluate the connectivity parameters alpha, beta, and gamma (alpha: across-domain (same class, different domain) connectivity, beta: across-class (same domain, different class) connectivity, gamma: across-both (different domain, different class) connectivity) for iWildCam and AstroClassification. We use the multi-crop augmentation strategy used in SwAV contrastive pretraining for iWildCam and the masking augmentation used in AstroClassification pretraining, respectively. Shen et al., 2022 requires that min(alpha, beta) > gamma for contrastive pretraining to work as a UDA method. We find that beta (across-class) < gamma (across-both) for iWildCam, which violates Shen et al.,’s assumption, whereas the assumption holds for AstroClassification. We also observe that both across-domain connectivity and across-class connectivity are much higher for AstroClassification compared to iWildCam, verifying our reasoning for the differing OOD downstream performance with pretraining.
> | Dataset             | across-domain | across-class | across-both  |
> |---|---:|---:|---:|
> | iWildCam            | 0.116         | 0.071        | 0.076        |
> | AstroClassification | 0.287         | 0.159        | 0.097        |
>
> [1] Connect, Not Collapse: Explaining Contrastive Learning for Unsupervised Domain Adaptation, 2022.

---

> > ### Author Response · Authors · 2023-11-19
> > **Response (2/2)**
> >
> > > From Table 2, it seems like the results are sensitive to the target augmentations. For iwilds most of the gains seem to come from the targeted augmentations. But for the astro datasets the ID test acc for astroclassification is better without targeted augmentations. On a new task or dataset, how should one choose the targeted augmentations?
> >
> > **We describe a general procedure for designing targeted augmentations from the source and target data distributions in Section 4 of the paper**, as well as an example of designing the redshifting augmentation used for AstroClassification within this framework. Connect Later improves OOD accuracy on all datasets despite the varying effectiveness of targeted augmentations.
> >
> > > Test-time methods, like test-time augmentation/training also makes use of unlabelled data. These methods aim to adapt to the distribution shift as they occur. Furthermore, some of these pre-training objectives like masked auto-encoding have been used for test-time adaptation [2]. It may be beneficial to pose these methods wrt Connect Later.
> >
> > Test-time methods are complementary to Connect Later – Connect Later can train a better base model for test-time training. Overall, test-time training works in a different online setting where the unlabeled/test example is only available before we must make a prediction on it. We will add this to the related work - we thank the reviewer for this suggestion.
> >
> > > Sec 4 gives the technical details for creating augmentations before explaining the tasks. It may not be clear what the tasks for AstroClassification and Redshift are about and so the technical details are not easy to understand. I would suggest moving Sec 5 before 4.
> >
> > We thank the reviewer for this suggestion and will make the change in the final version.
> >
> > > In Fig 2, top row, what do the colors mean? Why does augmentation introduce errors?
> >
> > Each color represents a set of flux measurements taken at a different wavelength. The redshifting augmentation increases the size of the error bars because this augmentation simulates placing each source object further away (i.e. at a higher redshift). Objects that are further away appear less bright and are not able to be resolved clearly (faraway objects look small and blurry), leading to larger uncertainties on each flux measurement. This is by design, since the target dataset objects look more like this (see top row, right of Fig 2).
> >
> > > It may be useful to have additional columns in the main table of results (e.g. Tab 2), that describes what the different methods are. E.g., whether finetuning augmentations are used, and with checks in the rows for the relevant methods.
> >
> > We thank the reviewer for this suggestion and will make the change in the camera-ready version.

---

> ### Author Response · Authors · 2023-11-21
>
> Hello!
>
> We were wondering if you've had a chance to look over our comments? We show that the choice of pretraining method is not the cause for inconsistent results by testing with SwAV contrastive pretraining on another benchmark dataset, Camelyon17-WILDS, where pretraining does improve over ERM. We also show that Connect Later sets a new state-of-the-art on Camelyon17-WILDS with DenseNet-121. There are also other experiments and details in the comments above!
>
> Please let us know if you have other questions or concerns. Thank you!

---

> ### Comment · Reviewer_atYU · 2023-11-22
>
> Thanks for the clarifications and discussions. My biggest concern is still on the generality of the method, e.g. how would one use the proposed method on a new dataset? It seems like different datasets uses different pre-training methods/only considers one pre-training method and the target augmentations need to be carefully designed. Thus, I am inclined to keep my score.

---

### Official Review · Reviewer_vvbp · 2023-11-02

**Soundness:** 1 poor
**Presentation:** 3 good
**Contribution:** 2 fair
**Rating:** 3
**Confidence:** 4

**Summary:**

This manuscript deals with the domain adaptation problem, i.e., the setting in which there is a domain shift between a source and target domain and where only unlabelled data is available for the target domain. One approach to this problem is to use self-supervised learning using both the source and target domain data, followed by fine-tuning with the labelled source domain data.

The authors begin with the observation that this approach leads to inconsistent results compared to regular supervised training (ERM). They hypothesize this is because the self-supervised methods learn features that are strongly domain-specific, which means that when the model is fine-tuned the classifier from the last layer might not generalize beyond the source distribution. The authors propose a solution which involves using targeted augmentations (i.e., augmentations that are specifically designed to remove spurious domain-dependent features) during the fine-tuning phase. They show that this improves performance on three different datasets.

**Strengths:**

This paper looks at an interesting problem: Unsupervised domain adaptation is a very relevant problem (e.g., in many cases practitioners have labelled datasets available from a curated/lab setting, but need to generalize to actual deployment conditions). At the same time, self-supervised learning has become very popular, but its performance in the context of unsupervised domain adaptation (UDA) is still unclear.

This paper introduces a new dataset (RedShifts), and contains some encouraging results on two different domains (camera trips and astronomical observations) while performing several relevant ablations (model scale, whether to train the last layer before fine-tuning, strength of pre-training augmentations).

The paper is relatively easy to read, with good writing and a clear structure.

**Weaknesses:**

Overall, I find this paper lacking in a variety of areas:

The authors' refer several times to Shen et al. (2022). This paper argues that contrastive pre-training is beneficial as long as data augmentations ensure that augmented examples must be more likely to change class or domain than changing both. The authors of this manuscript seem to argue in section 3 that the bad results they observe are explained by a violation of this assumption from Shen et al. (2022). However, this isn't tested rigorously (see section 6 of Shen et al., 2022, for examples on how to evaluate connectivity on real world datasets).

A second shortcoming is that different self-supervised methods were used for the iWildCam-WILDS dataset (SWaV contrastive learning) and the astronomical time-series (masked autoencoding). This makes it hard to draw conclusions from table 1, since it is unclear which differences can be attributed to the use of different pre-training objectives.

The results on the iWildCam dataset are not very strong (given the means and standard deviations listed, there's a 20% chance that ERM + targeted augs outperforms Connect Later for any run). This leaves the improvements in the astronomical time-series datasets as the stronger proof of Connect Later's performance. But these results are a bit confusing to me: How is it that standard fine-tuning improves in-domain performance (tables 1 and 2)? This suggests to me that these datasets are possibly very small and benefit strongly from the regularization that pre-training provides. In that case, is Connect Later really addressing a domain shift issue? Or is it just addressing a regular overfitting issue? The latter seems plausible given the very small size of the dataset (6,74 objects). I would argue that in order to draw conclusions, the method should be tested on a wider variety of larger datasets.

A baseline that seems to be missing as well is the use of general augmentations during fine-tuning (my understanding is that standard fine-tuning uses no augmentations at all). This would clarify whether it is important to use targeted augmentations during fine-tuning, or whether any form of augmentations would be fine.

Overall, I find the insights that this paper provides limited: It is not surprising to me that targeted augmentations would help with fine-tuning, given that they were shown to help with regular ERM. The bigger question to me is the one raised in section 3: Why does pre-training not always lead to increased OOD performance? This question remains largely unanswered. I think this paper would be better if it provided, for example, (1) clear evidence that the assumptions from Shen et al. (2022) are routinely violated in real-world datasets and commonly used augmentations, (2) a thorough evaluation of how different pre-training techniques (MAEs, contrastive learning) perform in the UDA setting, or (3) strong evidence that targeted augmentations are particularly important compared to general augmentations.

**Questions:**

See weaknesses.

---

> ### Author Response · Authors · 2023-11-19
> **Response (1/2)**
>
> We thank the reviewer for the feedback. vvbp notes that the paper **looks at an interesting problem** and contains **encouraging results on two different domains**. We answer specific questions below:
>
> > Why does pre-training not always lead to increased OOD performance? This question remains largely unanswered. I think this paper would be better if it provided, for example, (1) clear evidence that the assumptions from Shen et al. (2022) are routinely violated in real-world datasets and commonly used augmentations
>
> **We empirically evaluated connectivity during the rebuttal period and show that the connectivity measurements of iWildCam-WILDS violates the assumptions from Shen et al, 2022 that govern the success of contrastive pretraining for domain adaptation, while AstroClassification has much higher connectivity** (details below). This validates our intuition that generic augmentations, such as the ones used in SwAV contrastive pretraining, are not enough to connect the domains for certain datasets and shifts.
>
> Specifically, we evaluate the connectivity parameters alpha, beta, and gamma (alpha: across-domain (same class, different domain) connectivity, beta: across-class (same domain, different class) connectivity, gamma: across-both (different domain, different class) connectivity) for iWildCam and AstroClassification.
> We use the multi-crop augmentation strategy used in SwAV contrastive pretraining for iWildCam and the masking augmentation used in AstroClassification pretraining, respectively. Shen et al., 2022 requires that min(alpha, beta) > gamma for contrastive pretraining to work as a UDA method. **We find that beta (across-class) < gamma (across-both) for iWildCam, which violates Shen et al.,’s assumption, whereas the assumption holds for AstroClassification. We also observe that both across-domain connectivity and across-class connectivity are much higher for AstroClassification compared to iWildCam, verifying our reasoning for the differing OOD downstream performance with pretraining.** We thank the reviewer for this suggestion and will add this to the paper in the final revision.
>
> | Dataset             | across-domain | across-class | across-both  |
> |---|---:|---:|---:|
> | iWildCam            | 0.116         | 0.071        | 0.076        |
> | AstroClassification | 0.287         | 0.159        | 0.097        |
>
> We follow Shen et al., 2022 to measure connectivity. First, we train a binary classifier to distinguish between class-domain pairs (e.g. for across-domain connectivity, the classifier distinguishes between augmented examples with the same class but different domain). We compute the connectivity as the test 0-1 error of these classifiers (1 - test_accuracy), averaged over class-domain pairs.
>
> > different self-supervised methods were used for the iWildCam-WILDS dataset (SWaV contrastive learning) and the astronomical time-series (masked autoencoding). This makes it hard to draw conclusions from table 1, since it is unclear which differences can be attributed to the use of different pre-training objectives.
>
> **During the rebuttal period, we ran the standard fine-tuning baseline with SWaV contrastive pretraining on another benchmark, Camelyon17-WILDS, and found that the choice of pretraining method is not the source of inconsistent OOD results from pretraining.** On Camelyon17-WILDS, contrastive pretraining with standard fine-tuning improves accuracy over both ERM+targeted augs and ERM. This is very different from iWildCam-WILDS, where contrastive pretraining with standard fine-tuning did not even improve accuracy over ERM. Our results on Camelyon17-WILDS and iWildCam-WILDS both use SwaV contrastive learning, showing that the choice of pretraining method is not the source of inconsistent OOD results. **On Camelyon17-WILDS, Connect Later also achieves the new state-of-the-art ID and OOD accuracy.**
>
> |                      | ID Val Avg Acc | OOD Test Avg Acc |
> |---|---:|---:|
> | ERM                  | 89.3 (0.9)     | 65.2 (1.1)       |
> | Standard fine-tuning | 92.3 (0.2)     | 91.4 (0.9)       |
> | ERM + targeted augs  | 96.7 (0.0)     | 90.5 (0.4)       |
> | DANN                 | 86.1 (1.3)     | 64.5 (1.2)       |
> | CORAL                | 92.3 (0.7)     | 62.3 (1.9)       |
> | ICON          | 90.1 (0.3)     | 93.8 (0.2)       |
> | Connect Later        | **98.5 (0.0)**     | **94.9 (0.4)**       |
>
> DANN and CORAL results are from Sagawa et al., 2022 (10 seeds) and the previous state-of-the-art results are from ICON (https://github.com/a-tea-guy/ICON, 3 seeds). All other results are averaged over 20 seeds. We report 95% confidence intervals in parentheses.

---

> > ### Author Response · Authors · 2023-11-19
> > **Response (2/2)**
> >
> > > The results on the iWildCam dataset are not very strong (given the means and standard deviations listed, there's a 20% chance that ERM+targeted augs outperforms Connect Later for any run).
> >
> > We ran additional seeds for iWildCam to clarify these results, for a total of 15 seeds for each method. We report 95% confidence intervals on the mean estimate in parentheses and demonstrate that **the mean F1 results of ERM+targeted augs are outside of the 95% confidence intervals of Connect Later. Overall, Connect Later achieves the state-of-the-art OOD accuracy on iWildCam-WILDS (for ResNet-50), Camelyon17-WILDS (for DenseNet-121), and AstroClassification.**
> >
> > |                      | ID Test Macro F1 | OOD Test Macro F1 |
> > |---|---:|---:|
> > | ERM                  | 46.4 (0.5)       | 30.4 (0.6)        |
> > | Standard fine-tuning | 46.4 (0.8)       | 31.2 (0.6)        |
> > | ERM + targeted augs  | 51.4 (0.6)       | 36.1 (0.7)        |
> > | DANN                 | 48.5 (3.2)       | 31.9 (1.6)        |
> > | CORAL                | 40.5 (1.6)       | 27.9 (0.5)        |
> > | Noisy Student        | 47.5 (1.0)       | 32.1 (0.8)        |
> > | Connect Later        | 51.7 (0.8)       | 36.9 (0.7)        |
> >
> >
> >
> > > A baseline that seems to be missing as well is the use of general augmentations during fine-tuning (my understanding is that standard fine-tuning uses no augmentations at all). This would clarify whether it is important to use targeted augmentations during fine-tuning, or whether any form of augmentations would be fine.
> >
> > **During the rebuttal period, we ran this suggested baseline for AstroClassification and found that Connect Later outperforms standard fine-tuning+generic augmentations by 7% on OOD accuracy (79.9% vs. 72.7%).** The difference in ID accuracy between Connect Later and standard fine-tuning+generic augmentations is much smaller (80.54% vs 79.9%) in comparison.
> > This result demonstrates that targeted augmentations in Connect Later are key to OOD performance, and are not replaceable with generic augmentations. For the generic augmentations, we applied a 20% masking augmentation at fine-tuning time. We thank the reviewer for this suggestion and will add this to the paper in the final revision.
> >
> > > is Connect Later really addressing a domain shift issue? Or is it just addressing a regular overfitting issue? The latter seems plausible given the very small size of the dataset (6,74 objects). I would argue that in order to draw conclusions, the method should be tested on a wider variety of larger datasets.
> >
> > While pretraining could have a regularization effect for small datasets, **we tested Connect Later on two larger distribution shift benchmarks: iWildCam-WILDS (130k labeled examples) and Camelyon17-WILDS (300k labeled examples), where Connect Later still improves OOD performance**. We also tested standard fine-tuning with generic augmentations on AstroClassification (as a generic form of regularization), and this underperforms Connect Later by 7% OOD.

---

> > > ### Author Response · Authors · 2023-11-21
> > >
> > > Hello!
> > >
> > > We were wondering if you've had a chance to look over our comments? We implemented connectivity evaluation and found that iWildCam does in fact violate a Shen et al. assumption. We also show that the choice of pretraining method is not the cause for inconsistent results by testing with SwAV contrastive pretraining on another benchmark dataset, Camelyon17-WILDS, where pretraining does improve over ERM. Connect Later sets a new state-of-the-art on Camelyon17-WILDS with DenseNet-121. There are also other experiments and details in the comments above!
> > >
> > > Please let us know if you have other questions or concerns. Thank you!

---

### Author Response · Authors · 2023-11-21
**General response**

We thank all the reviewers for their thorough feedback. The reviewers note that **“this paper looks at an interesting [and] relevant problem”** and presents **”supportive results on several real-world datasets”**.

In response to the feedback, we added the following during the rebuttal period:
- **Empirical evaluation of connectivity**: we empirically measured connectivity following [1] and found that **connectivity on iWildCam is much lower than AstroClassification and even violates the theoretical condition** for contrastive pretraining to work for domain adaptation from [1]. This supports our connectivity-based explanation for why standard fine-tuning does not improve over ERM on iWildCAM-WILDS.
- **New SoTA results on Camelyon-WILDS benchmark**: We applied Connect Later to the Camelyon-WILDS benchmark. We found that Connect Later achieves a new SoTA on Camelyon-WILDS, achieving 94.9% OOD accuracy (1.1% above previous SoTA with unlabeled data).
- **Further evidence of inconsistent OOD accuracy from pretraining+standard fine-tuning**: On Camelyon-WILDS, we used the same pretraining method (SwAV contrastive pretraining) as iWildCam-WILDS. Standard fine-tuning on Camelyon-WILDS improves significantly over ERM while on iWildCam-WILDS it does not, supporting the finding that standard fine-tuning+pretraining brings inconsistent OOD results (contrary to [1]). In particular, **the inconsistency in pretraining results persists cannot be explained by the choice of pretraining method**.
- **Standard fine-tuning with generic augmentations is not a replacement for targeted augmentations**: On AstroClassification, standard fine-tuning with generic masking augmentations underperforms Connect Later by 7% OOD.

**Overall summary:**

- We find that **standard fine-tuning after pretraining does not always improve OOD accuracy** over ERM, contrary to [1]. While standard fine-tuning improves over ERM on Camelyon-WILDS, AstroClassification, and Redshifts, there is not a significant improvement in iWildCam-WILDS.

- We hypothesize that for such datasets (iWildCam-WILDS), the connectivity measures across domains and classes are too low for pretraining to produce transferable representations (from [1]). Empirical measures of connectivity show that indeed, **the connectivity on iWildCam is much lower than AstroClassification and even violates the theoretical conditions from [1]**.
- To address this, we introduce Connect Later, which uses targeted augmentations designed with knowledge of the distribution shift to improve connectivity at fine-tuning time. The goal of Connect Later is to 1) leverage pretrained representations within domains/classes even when connectivity across domains/classes is low, and  2) enhance the transferability of pretrained representations otherwise. We also provide a methodology for designing targeted augmentations for new datasets. We also give a simple theoretical example of when Connect Later improves OOD accuracy over standard fine-tuning and ERM when connectivity conditions are violated.

- **Connect Later sets a new state-of-the-art on three benchmarks**, improving OOD performance on AstroClassification by 3%, iWildCam-WILDS with ResNet-50 by 0.8%, and Camelyon17-WILDS with DenseNet121 by 1.1%. Connect Later also achieves the best performance on a new dataset we curate, Redshifts. Connect Later improves OOD performance on all datasets, regardless of connectivity.

[1] Shen, et al. Connect, Not Collapse: Explaining Contrastive Learning for Unsupervised Domain Adaptation, 2022.

---

### Meta-Review · Area_Chair_e28o · 2024-01-07

**Metareview:**

While this paper significantly improved during the revision process, in the end one reviewer decided to keep their recommendation to reject, and the other two reviewers did not participate in the discussion. In light of this, it is best for the paper to get a fresh set of eyes, which hopefully with the improvements made will lead to publication down the line.

**Justification For Why Not Higher Score:**

no votes to accept

**Justification For Why Not Lower Score:**

N/A

---

### Decision · Program_Chairs · 2024-01-16

Reject